# Medical Named Entity Extraction from Chinese Resident Admit Notes Using Character and Word Attention-Enhanced Neural Network

**DOI:** 10.3390/ijerph17051614

**Published:** 2020-03-02

**Authors:** Yan Gao, Yandong Wang, Patrick Wang, Lei Gu

**Affiliations:** 1School of Automation, Central South University, Changsha 410083, China; gaoyan@csu.edu.cn (Y.G.); wydxtu@163.com (Y.W.); llsyme@163.com (L.G.); 2PTY LTD., Changsha 410083, China

**Keywords:** Chinese electronic medical record, resident admit notes, attention mechanism, named entity recognition, neural network

## Abstract

The resident admit notes (RANs) in electronic medical records (EMRs) is first-hand information to study the patient’s condition. Medical entity extraction of RANs is an important task to get disease information for medical decision-making. For Chinese electronic medical records, each medical entity contains not only word information but also rich character information. Effective combination of words and characters is very important for medical entity extraction. We propose a medical entity recognition model based on a character and word attention-enhanced (CWAE) neural network for Chinese RANs. In our model, word embeddings and character-based embeddings are obtained through character-enhanced word embedding (CWE) model and Convolutional Neural Network (CNN) model. Then attention mechanism combines the character-based embeddings and word embeddings together, which significantly improves the expression ability of words. The new word embeddings obtained by the attention mechanism are taken as the input to bidirectional long short-term memory (BI-LSTM) and conditional random field (CRF) to extract entities. We extracted nine types of key medical entities from Chinese RANs and evaluated our model. The proposed method was compared with two traditional machine learning methods CRF, support vector machine (SVM), and the related deep learning models. The result shows that our model has better performance, and the result of our model reaches 94.44% in the F1-score.

## 1. Introduction

An electronic medical record (EMR) is a textual record of medical activities [1,2]. The development of information technology has promoted the growth of electronic medical records. At the same time, the value of medical information is becoming more and more important. Due to their unstructured nature, extracting information from clinical notes is very difficult [3]. Therefore, how to effectively extract patients’ disease information is the primary requirement for studying the causes, development, and evolution of patient morbidity. Entity extraction of EMR is the basis of studying patients’ disease. It has a wide range of application scenarios, such as medical information retrieval, question and answer system, clinical decision support, etc. Therefore, accurate extraction of medical entities is crucial to get and use medical information.

China, as a large population country, has produced more and more electronic medical records in recent years. Therefore, the effective use of Chinese EMR and the effective extraction of information from Chinese EMR are of great significance to public health. The resident admit note (RAN) is a type of electronic medical record, which contains a large number of descriptions of patients’ condition, and it is the first-hand information for studying the patient’s diseases. The information of original Chinese RANs is unstructured, so it is highly significant to accurately extract the entity information inside. Figure 1 shows part of one annotated Chinese RAN collected from a famous hospital in Hunan, China. Its main contents include the chief complaint, present disease, history of past disease, personal history, family history, and physical examination. The words marked in different colors represent different medical entities. For example, 身体部位 (body part) is marked in blue, 医学发现 (medical discovery) is marked in green, 疾病 (disease) is marked in red, and 治疗 (treatment) is marked in yellow, etc. Our task is to extract these important medical entities from the original Chinese RANs. Specific annotation information can be seen in Section 3.

Like biomedical named entity recognition (NER) in English, medical named entity recognition of Chinese text also has some challenges. Firstly, one entity can contain multiple words, which requires the NER system to identify the physical boundaries. For example, entity “各瓣膜听诊区 (valve auscultation area)” consists of words “各”, “瓣膜”, “听诊”, and “区”. Secondly, the same word can be an entity or part of an entity, such as the two entities “腹壁 (abdominal wall)” and “腹壁静脉 (abdominal wall vein)”, both containing “腹壁 (abdominal wall)”. Lastly, there are some abbreviations in medical texts, which will increase the difficulty of extracting entities. Besides, unlike English, every Chinese word has its own meaning, and every character in the word also has its own meaning. So, it is better to combine the information of characters and words. For example, “腹壁静脉 (abdominal wall vein)” can be divided into words “腹壁 (abdominal wall)” and “静脉 (vein)”, and the words “腹壁 (abdominal wall)” and “静脉 (vein)” can be divided into “腹”,“壁” and “静”, “脉”, respectively. We should consider the words “腹壁 (abdominal wall)”, “静脉 (vein)”, and characters “腹”, “壁”, “静”, “脉” for Chinese entity recognition at the same time.

In this study, for medical entity of Chinese RANs, we propose a medical entity recognition model based on character and word attention-enhanced (CWAE) neural network. Firstly, we obtain Chinese word embeddings and character-based embeddings through character-enhanced word embedding (CWE) and convolutional neural network (CNN). Then, we use the attention mechanism to weight the character-based embedding and word embedding together. Through this method, the new word embedding was obtained, which fully combined the information of characters and words. After that, the new word embedding is fed to the bidirectional long short-term memory (BI-LSTM). In this way, the context semantic information of the entities is obtained. Finally, BI-LSTM is combined with the conditional random field (CRF) to predict medical entities. We annotated nine types of medical entities on 355 RANs from a famous hospital in Hunan Province, China. They included 医学发现 (medical discovery), 时间词 (temporal word), 检查 (inspection), 检验 (laboratory test), 治疗 (treatment), 疾病 (disease), 药物 (medication), 身体部位 (body part), and 测量数据 (measurement) in these RANs. Additionally, we comparatively evaluated our model on these nine types of entities. The result shows that our model has a better performance, and the result of our model reached 94.44% in the F1-score.

There are three main contributions in this paper:For the characteristics of Chinese RANs, we propose use of the attention mechanism to combine character and word information, which further improves the expression ability of words.We annotated nine types of entities on Chinese RANs, including medical discovery, temporal word, inspection, laboratory test, treatment, disease, drug, body part, and measurement.We achieved an F1-score of 95.93% for medical discovery; an F1-score of 86.83% for temporal word; an F1-score of 94.61% for inspection; an F1-score of 83.54% for laboratory test; an F1-score of 87.48% for treatment; an F1-score of 89.56% for disease; an F1-score of 78.82% for medication; an F1-score of 97.02% for body part; an F1-score of 94.73% for measurement; and an F1-score 94.44 for all the medical entities.

The remainder of the paper is organized as follows: Section 2 introduces the related works of entity extraction. Section 3 focuses on the detailed description of the experimental dataset, medical named entity recognition tasks, and the detailed introduction of our model. In Section 4, the experimental results and analysis are provided. Finally, Section 5 addresses our conclusion and future work. 

## 2. Related Work

With the advent of the medical big data era, more and more attention has been paid to knowledge mining and the utilization of electronic medical records. The information extraction from clinical free text is the most fundamental task [3]. The medical entity is the carrier of important information, and extracting medical entity is very important to public health. In recent years, with the rapid development of deep learning and natural language processing technology, information extraction technology is becoming more and more mature. More and more models are built to process biomedical texts.

There are many mature methods for biomedical text entities in English. The methods of named entity recognition mainly include traditional machine learning and deep learning. Traditional machine learning mainly includes logistic regression (LR), support vector machines (SVMs), hidden Markov model (HMM), and CRF, etc. Li et al. [4] used the artificial feature-based CRF for gene entity recognition, and the F1 score was 87.28% on the Biocreative II GM corpus. The Biocreative II GM corpus is provided by the gene mention (GM) tagging task, and the task is concerned with the named entity extraction of gene and gene product mentions in text. Wang et al. [5] used the SVM for biomedical named entity recognition on the JNLPBA corpus. The JNLPBA corpus is provided by the BioNLP/JNLPBA Shared Task 2004, and the task was organized by GENIA Project. Wang et al. used a lot of artificial features, including local features, full-text features, and external resource features. The F1 score was 71.7%. However, most of those methods rely on feature engineering, which is labor intensive. Deep learning does not require manual features, so it is becoming popular. Yao et al. [6] proposed a biomedical named entity recognition (Bio-NER) method based on deep neural network architecture, which has multiple layers. In their model, CNN can extract the sentence-level features, but it cannot extract the dependency features between characters in sentences. In order to efficiently make use of the sequence features of sentences, Li et al. [7] constructed BI-LSTM for entity recognition. They constructed the twin word embeddings and the sentence vector to enrich the input information. The F1 score was 88.6% on the Biocreative II GM corpus. Habibi et al. [8] proposed BI-LSTM-CRF with word embedding to improve biomedical named entity recognition. They evaluated their model using 24 corpuses covering five entity types. The average value of the F1 scores was 81.11%. For the detection of word-level and character-level features, Chiu et al. [9] constructed a BI-LSTM-CNNs model named the entity recognition model, which achieved a 91.62% F1 score on the CoNLL-2003 proprietary corpus and 86.28% F1 score on the OntoNotes corpus. Li et al. [10] proposed a CNN-BILSTM-CRF neural network model. They used CNN to train the character-level representation of words, then combined them with the word vectors obtained from large-scale corpus training, and then sent the combined word vectors to the BLSTM-CRF neural network for training. The F1 score on the Biocreative II GM corpus was 89.09% but only 74.40% on the JNLPBA corpus. To use widely available unlabeled text data for improving the performance of NER models, Sachan et al. [11] proposed the effective use of bidirectional language modeling for medical named entity recognition. They trained a bidirectional language model (Bi-LM) on unlabeled data and transferred its weights to an NER model with the same architecture as the Bi-LM. The best F1 score on corpus clinical notes is 86.11%. To pay attention to the significant areas when capturing features, Wei et al. [12] proposed an attention-based BILSTM-CRF model, and their model obtained an F1-score of 73.50% on JNLPBA corpus.

For Chinese named entity recognition, Ouyang et al. [13] proposed named entity recognition based on the BI-LSTM neural network with additional features. The experimental results showed that the BI-LSTM with word embedding trained on a large corpus achieved the highest F1 score of 92.47%. However, they did not combine CRF and consider the character information in words. Xiang et al. [14] proposed a Chinese NER method based on character-word mixed embedding (CWME). They averaged the word embedding and character embedding when the word contains only one character. Yang et al. [15] proposed deep neural networks for medical entity recognition in Chinese online health consultations. They utilized BI-LSTM-CRF as the basic architecture, and concatenated character embedding and context word embedding to learn effective features. 

Different from the methods mentioned above, we proposed a medical entity recognition model based on character and word attention-enhanced neural network for Chinese RANs. We initialized the embedding layer through the CWE model to obtain the word embedding and the character embedding. We used the attention mechanism to combine the information of the character and word. Then, through training, the model combined the information of characters and words with the best weight. In our experiment, we used 355 Chinese RANs from a famous hospital in Hunan Province, China. We annotated 医学发现 (medical discovery), 时间词 (temporal word), 检查 (inspection), 检验 (laboratory test), 治疗 (treatment), 疾病 (disease), 药物 (medication), 身体部位 (body part), and 测量数据 (measurement) in these RANs. Additionally, we evaluated our model on these nine entities. 

## 3. Materials and Methods 

### 3.1. Dataset

#### 3.1.1. Entity Annotation

The experimental data of this work were 355 RANs provided by a famous hospital in Hunan Province, China. The RANs contain lots of medical information. In order to increase the diversity, th-ese RANs came from six departments: Gastroenterology (9%), gynecology (4%), urology (8%), pediatrics (27%), nephrology (12%), and neurology (22%). Figure 2 shows the distribution of the sample. We can see that the source of the RANs is more extensive, and it covers seven patient groups.

In order to accurately extract Chinese EMR information and verify our extraction model, we invited relevant doctors to develop a medical entity labeling process. According to the characteristics of RANs, we took the doctor’s advice into consideration and developed nine types of medical entities: 医学发现 (medical discovery), 时间词 (temporal word), 检查 (inspection), 检验 (laboratory test), 治疗 (treatment), 疾病 (disease), 药物 (medication), 身体部位 (body part), and 测量数据 (measurement). The annotation process of the dataset is shown in Figure 3. First of all, according to the doctor’s advice and the data annotator, we made the annotation rules to provide the basis for the subsequent entity annotation. Secondly, according to the annotation rules, we used the text annotation tool to annotate nine types of entities, and the annotation tool was e-host. Finally, we tested the rationality of the entities we annotated by the consistency test, and transformed the data into a training set and test set.

In the end, we annotated 66943 medical entities and obtained 25,9074 words. Table 1 shows the statistics for all samples in the experiment, including the statistics for all sentences, words, and characters in the sample. Table 2 shows the statistics for all the entities, including the number and proportion of each type of entity.

#### 3.1.2. Consistency Check

Three labeling staff and a neurology professional doctor participated in the labeling work. This work was three rounds of pre-labeling work, marking 5, 20, and 50 RANs in the three rounds, respectively. We performed consistency checks on the annotation results of each round. The calculation formulas are Formulas (1)–(3):(1)$$P=\frac{\mathrm{The}\;\mathrm{number}\;\mathrm{of}\;\mathrm{consistent}\;\mathrm{labels}\;\mathrm{marked}\;\mathrm{by}\;A\;\mathrm{and}\;B}{\mathrm{The}\;\mathrm{number}\;\mathrm{of}\;\mathrm{labels}\;\mathrm{marked}\;\mathrm{by}\;B},$$
(2)$$R=\frac{\mathrm{The}\;\mathrm{number}\;\mathrm{of}\;\mathrm{consistent}\;\mathrm{labels}\;\mathrm{marked}\;\mathrm{by}\;A\;\mathrm{and}\;B}{\mathrm{The}\;\mathrm{number}\;\mathrm{of}\;\mathrm{labels}\;\mathrm{marked}\;\mathrm{by}\;A\;},$$
(3)$$F=\frac{P\times R\times 2}{P+R},$$
where A and B are the labelers, *P* is the precision, and *R* is the recall rate. *P* represents the proportion of the same entities annotated by A and B to the entities marked by B. Instead, *R* represents the proportion of the same entities annotated by A and B to the entities labeled by A. *F* represents the harmonic mean of *P* and *R*, which is used to indicate the label consistency. A larger *F* indicates higher entity consistency between A and B. The entity consistency statistics are shown in Figure 4. The *F* value of our data reaches 97.73%. From the results, we can see that the dataset we annotated is scientific and accurate.

### 3.2. Methods

In this study, we transformed the named entity recognition (NER) task into sequence classification. The correct labels for a sentence were marked using the “BIO” Marking Standard [16], where “O” means that the word is not in the entity, “B” means that the word is the beginning of the entity, and “I” means that the word is inside the entity but not at the beginning. Each entity can be represented as “B-entity”, “I-entity”. Words that are not entities are represented as “O”. In our task, there were nine types of entities, so there were a total of 19 labels; they were: B-医学发现, I-医学发现, B-时间词, I-时间词, B-检查,I-检查, B-检验, I-检验, B-药物, I-药物, B-治疗, I-治疗, B-测量数据, I-测量数据, B-疾病, I-疾病, B-身体部位, I-身体部位, and O. Table 3 gives an illustration of the BIO labeling. Firstly, we split the electronic text into sentences, then used the jieba word segmentation tool to segment the sentences into words. Finally, we split each word into multiple characters, and used special PAD markers to make the characters of all words have the same length.

With the development of deep learning technology, deep learning shows superior ability for sequence classification tasks [17,18]. Considering the characteristics of Chinese electronic medical records, we proposed a medical entity recognition model based on character and word attention-enhanced neural network. Figure 5 is the architecture of the model.

Our model has five layers, including word embedding by CWE, character-based embedding by CNN, attention mechanism, BI-LSTM layer, and CRF layer. The word embedding and character-based embedding are obtained through the CWE and CNN model, and the information of the words and characters is combined by the attention mechanism to get new word embedding. The BI-LSTM layer is used to learn the context of the sentence. The softmax function normalizes the output of BI-LSTM between 0 and 1. The CRF layer outputs the label sequence with the maximum score calculated.

#### 3.2.1. Word Embedding by CWE

Word and character embeddings mean that the words and characters are mapped to vectors. This work uses CWE [19] to train word and character embedding. CWE aims at predicting the target word, given context words and characters in a sliding window. For a word sequence $D=\left\{d_{1},\ldots ,d_{M}\right\}$, the objective of CWE is to maximize the average log probability:(4)$$L(D)=\frac{1}{M}\sum_{i=K}^{M-K}\log P_{r}(d_{i}|d_{i-K},\ldots ,d_{i+K}),$$
where K is the context window size of the target word. For the Chinese character set C and the Chinese word set W, each character $c_{i}\in C$ is represented by vector $c_{i}$, and each word $w_{i}\in W$ is represented by vector $w_{i}$. CWE learns to maximize the average log probability in Formula (4) with a word sequence D. Formally, a context word $X_{i}$ is represented as:(5)$$X_{i}=\frac{1}{2}(w_{i}+\frac{1}{N_{i}}\sum_{m=1}^{N_{j}}c_{m}),$$
where $w_{i}$ is the word vector of the i-th word. $c_{m}$ is the m-th character vector in $d_{i}$, and $N_{i}$ is the number of characters. Through this model’s training, we finally get the word embedding and character embedding $c_{m}$.

#### 3.2.2. Character-Based Embedding by CNN

Convolutional neural network (CNN) have been used for many English sequence annotation tasks [9,20] to extract character information. CNN has the greatest impact on the field of health informatics [21]. Our model extracts the character features in word by CNN. Figure 6 shows the process of character-level CNN. The process of character-level CNN mainly includes two parts: Convolution and max-pooling. The specific calculation formula is as follows:(6)$$C_{i}^{k}=f(W^{k}*c\;[:,m+s-1]+b).$$

Let $L_{c}$ be the length of the character embedding and $W_{l}$ be the maximum length of a word in a mini-batch. $c\in R^{W_{l}\times L_{c}}$ is the character embedding in word $d_{i}$, which is calculated by the CWE model. b is the bias, ∗ is the dot product operator, s is the convolution stride, and f is the activation function, such as relu. Filters of different strides are used to compute the character-level features. k is the number of filters and $W^{k}$ is the k-th convolution weight. $C_{i}^{k}$ is the vector after convolution of the k-th filter. We can obtain a single feature of the word by max pooling of $C_{i}^{k}$. Finally, all the features are concatenated to obtain character-based embedding $C_{i}$.

We take Figure 6 as an example. In Figure 6, “静脉曲张” has four characters, k is set as 3, that is, there are three convolution kernels. s is set to 2, 3, and 4. By convolution and max pooling, three character features are obtained. Finally, we concatenate the character features as the character-based embedding. 

#### 3.2.3. Character and Word Attention Mechanism

The mechanism of attention stems from the study of human vision. In cognitive science, because of the bottleneck of information processing, humans selectively focus on all of the information while ignoring other visible information. For entity extraction, word information and character information in entities are both important, so we consider that our model can pay attention to both word and character information. 

We used the attention mechanism method to combine word and character. The union of the word and character is adds character information into word embedding, which is very effective for improving the expression ability of words. We propose the attention mechanism to combine the information of characters and words by referring to the study of Yang et al. [22]. Figure 7 shows the process of the character and word attention mechanism.

We concatenate character-based embedding and word embedding and get $\left[X_{i};C_{i}\right]$. Then, we transform $\left[X_{i};C_{i}\right]$ through the weight matrix and obtain the output through the activation function. Their calculation formula is as follows:(7)$${\bar{O}}_{g}=W_{g}[X_{i};C_{i}]+b_{g},$$
(8)$$O_{g}=\tanh({\bar{O}}_{g}),$$
where $W_{g}$ is the weight matrix in Formula (7), the input vector $X_{i}$ is the word vector of $d_{i}$, $C_{i}$ is the character-based embedding after CNN, and $b_{g}$ is the offset vector. tanh is the activation function:(9)$$z=U*O_{g}+b_{z},$$
(10)$$z=sigmoid(\bar{z}).$$

In Formulas (9) and (10), U is the weight matrix, and $b_{z}$ is the offset vector. sigmoid is the activation function. The output function is controlled between 0 and 1 by the activation function, and then the attention weight z is obtained. Finally, the word information and the character information are borrowed by the weight z. The calculation formula is as follows:(11)$${\bar{X}}_{i}=z*X_{i}+(1-z)C_{i}.$$

#### 3.2.4. BI-LSTM Layer

The bidirectional long-term memory neural network [23] is widely used in the natural language process. It can effectively deal with the sequence information and solve the problem of gradient disappearance.

For a given sentence $D=\left\{d_{1},\ldots ,d_{M}\right\}$, its new word embedding $\left({\bar{X}}_{1},\ldots ,{\bar{X}}_{M}\right)$ is the input of BI-LSTM. After the calculation of forward LSTM, we can get the hidden layer output ${\vec{h}}_{i}$ of each word. In the same way, after the calculation of the backward LSTM, we can get the hidden layer output ${\overset{\leftarrow }{h}}_{i}$ of each word. By concatenating, we can get the new output $h_{i}=[{\vec{h}}_{i};{\overset{\leftarrow }{h}}_{i}]$ of the words. Firstly, $h_{i}$ is prevented from overfitting by dropout. Then, we get the 19-dimensional vector by dimensionality reduction. Finally, the score of each tag is calculated by the softmax function.

#### 3.2.5. CRF Layer

CRF is commonly used in the field of natural language processing (NLP). It can take into account the context of the text and predict the tag of the sequence. We used CRF as the output layer of the neural network [24]. CRF can obtain the loss probability by predicting the label and get the transition probability between different labels.

For an input sentence $\bar{X}=\left\{{\bar{X}}_{1},\ldots ,{\bar{X}}_{M}\right\}$, its corresponding label is $y=\{y_{1}......,y_{M}\}$.$P\in R^{M\times T}$ is the output calculated by BI-LSTM and the softmax function, T is the number of labels (in this paper, the number of labels is 19, so T was set to 19). $p_{i,j}\in P$ represents the score of the i-th word and j-th tag in the sentence. Then, the final score is as follows:(12)$$s(\bar{X},y)=\sum_{i=1}^{n-1}A_{{}_{y_{i}y_{i+1}}}+\sum_{i=1}^{n}p_{i,y_{i}},$$
where $A\in R^{T\times T}$ is the transition probability matrix, which is obtained after training. The probability of the sequence y is generated as follows:(13)$$\log(p(y|\bar{X}))=s(\bar{X},y)-\log(\sum_{\bar{y}\in y_{X}}e^{s(X,\bar{y})}),$$
where $y_{X}$ represents all possible labels of sentence $\bar{X}$. Finally, the most probable tag sequence is calculated by Formula (14):(14)$$Y=\arg\max_{y\in y_{x}}s(\bar{X},y).$$

#### 3.2.6. Training Detail

The hyperparameters of our model are shown in Table 4. The word embeddings and character embeddings were trained by the CWE model and their dimensions were 100,100 respectively. For words that do not exist in the trained words (unregistered words), we randomly initialized them within $\left[-\sqrt{3/d\_w},+\sqrt{3/d\_w}\right]$ (d_w is the dimension of word embedding, and it was set to 100.). For unregistered characters, we randomly initialized them within $\left[-\sqrt{3/d\_c},+\sqrt{3/d\_c}\right]$ (d_c is the dimension of character embedding, and it was set to 100.). We used window 3, 4, 5 for char CNN filter and the output size of CNN was set to 200. The new word embedding size was set to 200 after CWAE. We used two-layer LSTM and set the state size of LSTM to 300.

The model was trained by a min-batch stochastic gradient descent (SGD) algorithm with a batch size of 20. Because SGD can better prevent redundancy. We used the Adam optimizer with default parameter settings for optimization. Adam optimizer is better than other methods according to the paper [25]. The epoch was set to 35, because we found that the model converged when the epoch was about 35. We used an initial learning rate of 0.001 and decayed the learning rate by multiplying it by 0.9 after every epoch. In addition, we used a gradient cut of 5.0 to reduce the gradient explosion by referring to the study of Pascanu et al. [26]. When the gradient explosion occurs, the gradient will be very large, which is bad for parameter learning. We used the gradient cut to control the gradient within 5.0 for better training. To avoid overfitting, we used dropout [27]. We selected the value of dropout by experiment, and it is better that the dropout is set to 0.5.

#### 3.2.7. Evaluation Metrics

We used three indicators to evaluate the results including *precision(P)*, *recall(R)*, and *F1-score(F1)*. Precision denotes the percentage of those correctly labelled in all forecast results. Recall denotes the percentage of correctly labelled positive results and overall positive results. The calculation formulas are as follows:(15)P=TP/(TP+FP),
(16)R=TP/(TP+FN),
(17)F1=2*P*R/(P+R),
where *FN* and *FP* represent the incorrect negative and positive predictions, and *TP* represents the correct positive predictions, which are actual correct predictions.

## 4. Results and Discussion

### 4.1. Result of Our Models

In the experiment, we used stratified sampling. We divided the data of each department into two parts, one part accounting for 70% as the training set and the other part accounting for 30% as the test set. Then, we put the training set of each department together as the final training set. Similarly, we put the test set of each department together as the final test set. Table 5 shows the recognition results of our model.

From the results of Table 5, we can see that the recognition results for different types of entities are different. This is normal because the number, length, and segmentation of every type of entity are different. The F1 values on 检验 (laboratory test) and 药物 (medication) are smaller than others. The F1 value on 检验 (laboratory test) is only 83.54%; the main reason is that the proportion of 检验 (laboratory test) in RAN is small. On RANs, different types of entities account for different proportions. From Table 4, we can see that the proportion of 检验 (laboratory test) is only 3.18%. For 药物 (drugs), in addition to a small proportion, inaccurate segmentation on entities also has an impact. For example, “泮托拉唑” is originally a whole, but it is divided into “泮”, “托拉”, “唑”, so it is hard to obtain the semantic information. However, our model can consider the combination of words and characters to improve the expressive ability of words, so the recognition result is improved according to the comparison in the following experiments.

### 4.2. Comparison with Traditional Models

We used two typical machine learning methods SVM and CRF for comparison and analysis.

CRF: We used the L-BFGS training algorithm (it is the default) with Elastic Net (L1 + L2) regularization. The maximum number of iterations was set to 100, which is the default value.SVM: We used the SVC function in sklearn. The penalty factor was set to 0.1, which is the default value. The number of features is large, so we used the linear kernel function [28] and max_iter was set to −1.

#### 4.2.1. Comparison of Overall

Table 6 shows the precision, recall rate, and F1 score of all the medical entities predicted by three ways. The results show that our model is both optimal in precision, recall, and F1 score, followed by CRF and SVM. The result of our model is 5.68% higher than CRF and 14.49% higher than SVM.

#### 4.2.2. Comparison of Each Category

We took medical discovery, temporal word, inspection, laboratory test, treatment, measurement, disease, medication, and body part as identification objects and experimented with SVM, CRF, and our model. The precision, recall, and F1 score of them are shown as Table 7. It can be known from the experimental results that the recognition of our method is the best on every type of entity. For medication, SVM predicts them with the highest precision, but because the recall rate is small, the F1 is low. Overall, the results of our model are better than CRF and SVM.

### 4.3. Comparison with Deep Learning Models

In order to test the effect of the model we built, we carried out an experiment to compare the performance of our model with the following deep learning models. Among them, these two models W-BILSTM and W-BILSTM-CRF did not combine character information, while the other models combined character information.

W-BILSTM: This model was used by Liubo Ouyang et al. [13]. They used word embedding as input, and fed them into a bidirectional long short-term for named entity recognition.W-BILSTM-CRF: We referred to the method proposed by Maryam Habibi et al. [8]. They used bidirectional LSTM and CRF as the basic model. The embedding is the input of the basic model.CWME-BILSTM-CRF: Xiang et al. [14] proposed the Chinese NER method based on character-word mixed embedding (CWME). The basic model is bidirectional LSTM and CRF, and CWME is the input of the basic model.CNN-BILSTM-CRF: This model was used by Li et al. [10]. They used CNN to train the character-level representation of words, then concatenated them with the word vectors.CWAM-BILSTM-CRF: This is the model proposed in this paper. We proposed a medical entity recognition model based on Character and Word Attention Enhanced (CWAM) neural network for Chinese RANs.CWAM-LSTM-CRF: We replaced BI-LSTM with LSTM in our model for comparison.

#### 4.3.1. Comparison of Overall

The overall recognition results are shown in Table 8. We know that our model is the best, which achieves precision of 94.16%, recall of 94.72%, and F1 of 94.44%. It is observed that W-BILSTM and W-BILSTM-CRF do not combine character information; the recognition results are poor, which are 4.99% and 2.56% less than our model, respectively. For Chinese RANs, the characters of every word have their own meaning. If the information of the character is neglected, the expression of the word becomes weaker.

For the combination of the word and character, the performance of our model is the best. The F1 value of CWME-BILSTM-CRF is 91.65%, which is 2.79% less than our model. The F1 value of CNN-BILSTM-CRF is 93.41%, which is 1.03% less than our model. It can be seen from the comparison that the character and word attention mechanism (CWAM) can better combine the information of characters and words, and improve the recognition result.

Besides, we changed our model’s BI-LSTM to LSTM, and the F1 score was reduced by 1.49%. For the unidirectional LSTM, only the forward sequence information of the sentence is considered, so the result is worse. Overall, our model absorbed the advantages of the above methods. It can better combine the information words and characters and improve the generalization ability of the model. It is a better model for medical entity recognition of Chinese RANs.

#### 4.3.2. Comparison of Each Category

For further comparison, we experimented with nine categories of entities separately. The final results of the experiment are shown in Table 9. It can be seen from the comparison that our model has the best F1 score other than “measurement” and “medication”. The method of CWAM-LSTM-CRF has the best recognition results for “measurement” and “medication”. Our model has an attention mechanism layer, and the number of parameters is relatively large. If there are fewer entities trained, the model will not be adequately trained, so the recognition result of our model on these two entities is not very good. However, our model performs better on other types of entities, and the overall recognition result is also the best.

### 4.4. Comparison of Convergence Rate

We monitored the change of the F-score and loss per epoch for the six deep learning models. The results are shown in Figure 8 and Figure 9. It can be seen from Figure 8 that our model can reach a higher F-score at the beginning of training, and the convergence speed is relatively fast. The F-score of CWAM-BILSTM-CRF and CNN-BILSTM-CRF is better than those of other deep learning models. As the number of epochs increases, the results of our model significantly exceed CNN-BILSTM-CRF. Additionally, we can see that our model has a fast convergence between 0–5 of epoch; when the epoch reaches 15, the model reaches stability. From Figure 9, it can be seen that the loss of CWAM-BILSTM-CRF reaches about 0.25 at the first epoch, which is lower than other models. For CWME-BILSTM-CRF, there is a large fluctuation in loss at about epoch 7, because there may be noise in the training data, which will affect the model. It indicates that the generalization ability of this model is poor.

## 5. Conclusions and Future Work

Medical named entity recognition is mature in foreign countries, but Chinese medical entity recognition is relatively late. For entity extraction of Chinese RANs, we proposed a medical entity recognition model based on character and word attention-enhanced neural network. Firstly, we obtained Chinese word embeddings and character-based embeddings through the CWE and CNN model. Then, the attention mechanism weighted the character-based embeddings and word embeddings together to get new word embeddings. The new word embeddings were fed to BI-LSTM and CRF to calculate the error for training. Finally, the trained model was used to predict the medical named entity. During the experiment, we used 355 RANs from a famous hospital in Hunan Province to annotate nine types of medical entities. To illustrate the superiority of our model, we compared our method with traditional machine learning methods (SVM, CRF). The experiments showed that the recognition results of our model are better than the traditional method, and the F1 score is the best. Then, we also experimented with related deep learning models to compare with our model. The results showed that the model we built has the better performance.

Our model also has some limitations. Our model mainly considers the semantic features of Chinese. English is different from Chinese, for example, a word in English can contain multiple Chinese characters, so this model is not suitable for English. However, for other languages, our model can be applied if the language expression is similar to Chinese. Besides, in our study, the word embeddings in the experiment were trained on 500 RANs, People’s Daily, and Wikipedia. The number of the BANs is not large, so there may be a lack of expression ability on word embeddings. In the next stage, we will strive to acquire more medical data and train character and word embeddings to further express words. The extraction of medical entities is only the first step of intelligent medicine. In-depth study of medical entity modification and the entity relationship is the goal of our next work. Besides, the RANs are the original data of the patient’s condition. Based on our current work, we will further study similar medical records and predict the preliminary diagnostic results.

## Figures and Tables

**Figure 1 ijerph-17-01614-f001:**
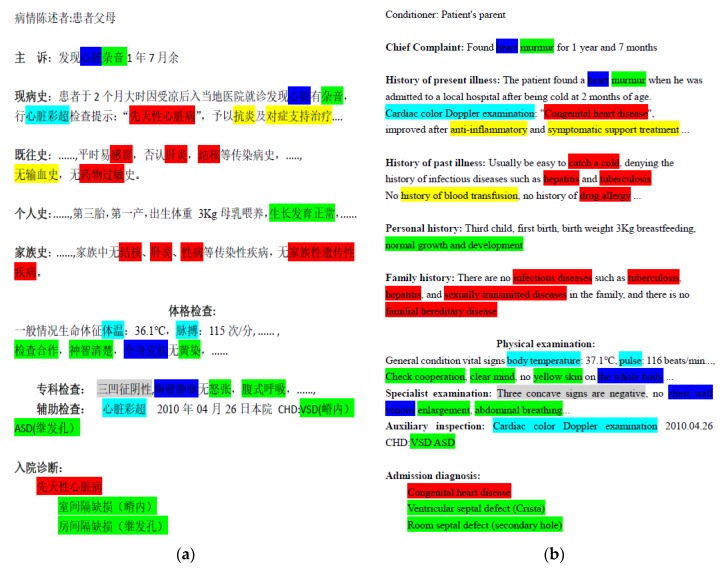
The example is the part of a resident admit note (RAN) collected from a famous hospital in Hunan. (**a**) Original RAN in Chinese and (**b**) its translation into English. The words of the text annotated in different colors represent different medical entities.

**Figure 2 ijerph-17-01614-f002:**
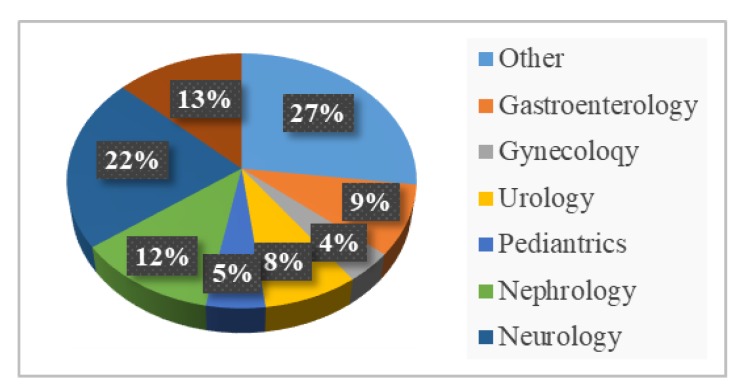
Distribution of the text of Chinese resident admit notes (RANs).

**Figure 3 ijerph-17-01614-f003:**
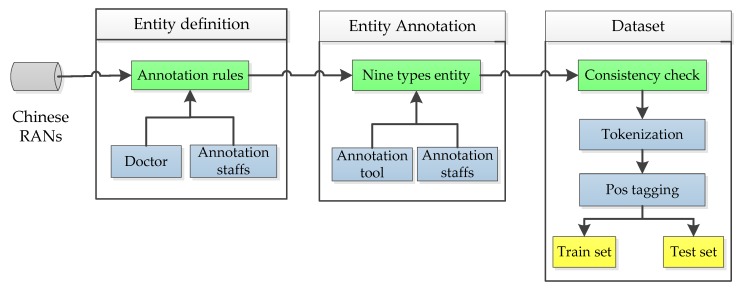
Annotation process.

**Figure 4 ijerph-17-01614-f004:**
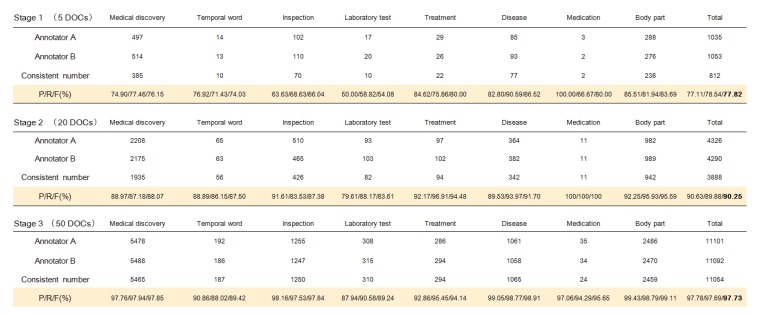
Stage 1 is the first round of statistics, stage 2 is the second round of statistics, and stage 3 is the third round of statistics. 5 documents (DOCs) represent 5 RANs, 20 DOCs represent 20 RANs, and 50 DOCs represent 50 RANs. DOCs means documents.

**Figure 5 ijerph-17-01614-f005:**
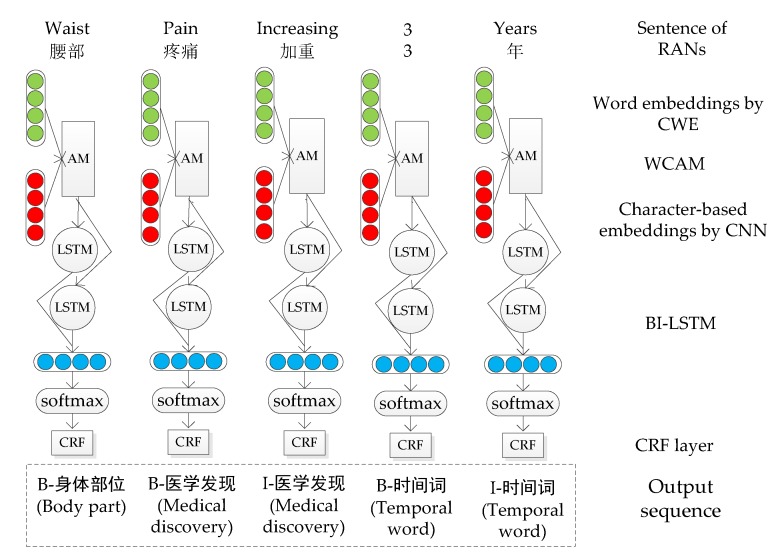
Architecture diagram of the medical entity recognition model based on character and word attention-enhanced neural network. AM: attention mechanism. The input is a sentence and the output is the label of each word in the sentence. CWAE: Character and word embedding attention mechanism.

**Figure 6 ijerph-17-01614-f006:**
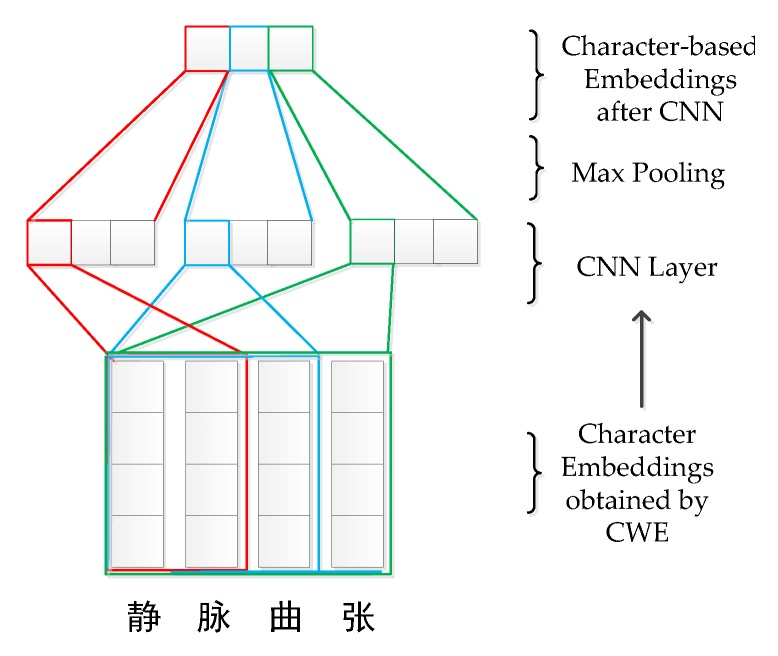
The process of the convolutional neural network (CNN) model. Character-based embedding is obtained by CNN of the four characters in word “静脉曲张”.

**Figure 7 ijerph-17-01614-f007:**
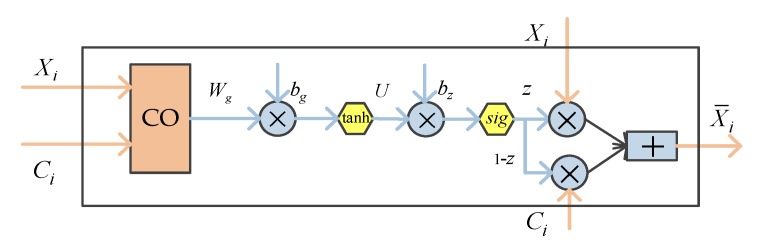
The process of the character and word embedding attention mechanism. In the figure, $X_{i}$ denotes the word vector. $C_{i}$ is character-based embedding obtained by convolution of character embeddings. “CO” means concatenation. $\bar{X}$ is the new word vector obtained by the character and word attention mechanism.

**Figure 8 ijerph-17-01614-f008:**
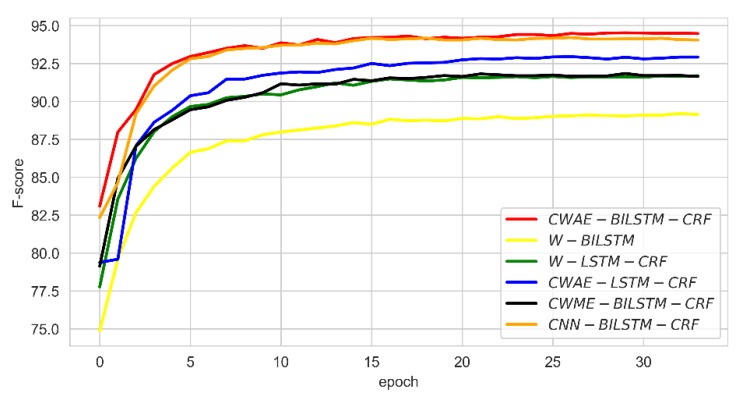
F-score of comparison result in tthe training process.

**Figure 9 ijerph-17-01614-f009:**
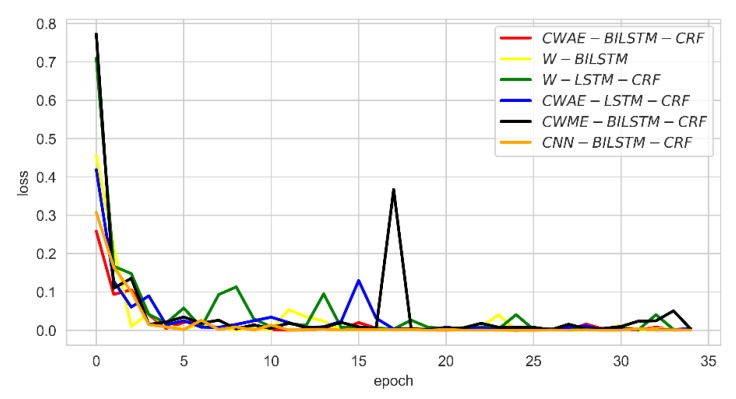
Loss of comparison result in the training process.

**Table 1 ijerph-17-01614-t001:** Statistics of all Chinese RANs.

	Sentences	Words	Features	Entities
Total	13,926	259,074	420,903	66,943
Average number of texts	54.61	1015.98	1650.6	262.52

**Table 2 ijerph-17-01614-t002:** Statistics of all Chinese medical entities.

Type of Entity	Count	Percentage
医学发现 (Medical discovery)	29,247	43.96%
时间词 (Temporal word)	1631	2.44%
检查 (Inspection)	6915	10.33%
检验 (Laboratory test)	2127	3.18%
治疗 (Treatment)	2601	3.88%
测量数据 (Measurement)	2839	4.24%
疾病 (Disease)	5286	7.90%
药物 (Medication)	1344	2.01%
身体部位 (Body part)	14,953	22.34%
Total	66,943	100%

**Table 3 ijerph-17-01614-t003:** Result of BIO labeling.

Words	English name	Characters	Labels	Words
患者	Patient	[患, 者]	O (Other)	患者
腰部	Waist	[腰, 部]	B-身体部位 (Body part)	腰部
疼痛	Pain	[疼, 痛]	B-医学发现 (Medical discovery)	疼痛
加重	Increasing	[加, 重]	I-医学发现 (Medical discovery)	加重
3	3	[3, PAD]	B-时间词 (Temporal word)	3
年	Years	[年,PAD,]	I-时间词 (Temporal word)	年

**Table 4 ijerph-17-01614-t004:** The parameters for our model.

Hyper-Parameters	Values
Word embedding size	100
Character embedding size	100
CNN filter sizes	2,3,4
CNN output size	200
New word embedding size	200
LSTM hidden size	300
Epochs	35
Dropout	0.5
Initial learning rate	0.001
Learning rate decay	0.9

**Table 5 ijerph-17-01614-t005:** The prediction results of our model.

Type of Entity	P	R	F1
Medical discovery	97.50	96.11	95.93
Temporal word	87.30	86.36	86.83
Inspection	95.06	94.17	94.61
Laboratory test	81.75	85.40	83.54
Treatment	85.60	89.43	87.48
Measurement	93.04	96.48	94.73
Disease	88.31	90.84	89.56
Medication	80.40	77.29	78.82
Body part	97.03	97.01	97.02
Total	94.16	94.72	94.44

**Table 6 ijerph-17-01614-t006:** Precision, recall, and F1 on all the medical entities with SVM, CRF, and our model.

Model	P	R	F1
SVM	83.37	81.45	82.40
CRF	89.70	87.84	88.76
Our model	**94.16**	**94.74**	**94.44**

**Table 7 ijerph-17-01614-t007:** The results comparison with traditional models on nine types of entities.

	**Medical Discovery**			**Temporal Word**			**Inspection**		
	**P**	**R**	**F1**	**P**	**R**	**F1**	**P**	**R**	**F1**
SVM	94.57	71.70	81.56	86.39	21.48	34.64	91.40	61.75	73.71
CRF	89.96	90.52	90.24	86.83	74.06	79.94	91.07	85.07	87.97
Our model	**95.75**	**96.11**	**95.98**	**87.30**	**86.36**	**86.83**	**95.06**	**94.17**	**94.61**
	**Laboratory test**			**Treatment**			**Measurement**		
	**P**	**R**	**F1**	**P**	**R**	**F1**	**P**	**R**	**F1**
SVM	76.68	52.99	62.67	83.83	58.95	69.22	84.91	31.60	46.06
CRF	67.80	68.81	68.30	75.46	58.13	65.67	89.37	87.42	88.38
Our model	**81.75**	**85.40**	**83.54**	**85.60**	**89.43**	**87.48**	**93.04**	**96.48**	**94.73**
	**Medication**			**Disease**			**Body part**		
	**P**	**R**	**F1**	**P**	**R**	**F1**	**P**	**R**	**F1**
SVM	87.37	20.34	33.00	87.32	53.61	66.43	89.53	66.52	76.26
CRF	71.62	51.21	59.72	78.34	72.08	75.08	92.58	93.22	92.90
Our model	80.40	**77.29**	**78.82**	**88.31**	**90.84**	**89.56**	**97.03**	**97.01**	**97.02**

Precision (P), recall (R), and F1 with SVM, CRF, and our model on medical discovery, temporal word, inspection, laboratory test, treatment, measurement, medication, disease, and body part.

**Table 8 ijerph-17-01614-t008:** Precision, recall, and F1 on all the medical entities with deep learning.

Models	P	R	F
W-BILSTM [13]	88.17	90.76	89.45
W-BILSTM-CRF [8]	91.88	91.87	91.88
CWME-BILSTM-CRF [14]	91.38	91.93	91.65
CNN-BILSTM-CRF [10]	93.98	92.85	93.41
CWAM-LSTM-CRF	92.67	93.22	92.95
CWAM-BILSTM-CRF	**94.16**	**94.72**	**94.44**

Note: W-BILSTM and W-BILSTM-CRF do not combine character information. Other models use different combination methods.

**Table 9 ijerph-17-01614-t009:** The results comparison with related deep learning models on nine types of entities.

	**Medical Discovery**			**Temporal Word**			**Inspection**		
	**P**	**R**	**F1**	**P**	**R**	**F1**	**P**	**R**	**F1**
W-BILSTM [13]	91.48	93.77	92.61	73.06	80.48	76.59	90.92	89.33	90.12
W-BILSTM-CRF [8]	94.45	94.74	94.59	83.51	82.62	83.06	92.80	90.15	91.45
CWME-BILSTM-CRF [14]	94.07	94.58	94.32	83.06	82.62	82.84	92.70	91.78	92.24
CNN-BILSTM-CRF [10]	95.91	95.75	95.83	85.95	83.42	84.67	95.12	93.24	94.17
CWAM-LSTM-CRF	**96.10**	95.50	94.87	84.53	81.82	83.15	93.67	93.24	93.45
CWAM-BILSTM-CRF	95.75	**96.11**	**95.98**	**87.30**	**86.36**	**86.83**	**95.75**	**96.11**	**94.61**
	**Laboratory**			**Treatment**			**Measurement**		
	**P**	**R**	**F1**	**P**	**R**	**F1**	**P**	**R**	**F1**
W-BILSTM [13]	73.13	77.48	75.24	71.50	82.11	76.44	83.60	88.09	85.78
W-BILSTM-CRF [8]	74.04	76.24	75.12	81.64	84.96	83.27	89.53	88.93	89.23
CWME-BILSTM-CRF [14]	73.78	78.71	76.17	80.66	83.94	82.27	90.00	92.11	91.04
CNN-BILSTM-CRF [10]	80.92	**85.79**	83.31	**87.05**	87.77	87.41	93.79	96.31	95.03
CWAM-LSTM-CRF	80.80	85.40	83.03	82.34	84.35	83.33	**94.26**	96.48	**95.36**
CWAM-BILSTM-CRF	**81.75**	85.40	**83.54**	85.60	**89.43**	**87.48**	93.04	**96.48**	94.73
	**Medication**			**Disease**			**Body part**		
	**P**	**R**	**F1**	**P**	**R**	**F1**	**P**	**R**	**F1**
W-BILSTM [13]	57.75	59.42	58.57	74.40	80.88	77.51	94.02	94.98	94.50
W-BILSTM-CRF [8]	70.21	63.77	66.84	81.48	87.70	84.48	95.04	95.13	95.08
CWME-BILSTM-CRF [14]	63.64	60.87	62.22	83.26.	83.03	83.15	94.54	95.18	94.86
CNN-BILSTM-CRF [10]	77.31	**80.68**	78.96	85.55	89.32	87.40	**97.04**	96.32	96.68
CWAM-LSTM-CRF	**82.41**	79.23	**80.79**	82.59	85.19	83.87	96.10	95.50	95.80
CWAM-BILSTM-CRF	80.40	77.29	78.82	**88.31**	**90.84**	**89.56**	97.03	**97.01**	**97.02**

Precision, recall, and F1 with existing neural network model and our model on Medical discovery, Temporal word Inspection, Laboratory test, Treatment, Measurement, Medication, Disease, Body part. W-BILSTM and W-BILSTM-CRF do not combine character information and other models use different combination methods.

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
