# Peer review of "Medical Named Entity Extraction from Chinese Resident Admit Notes Using Character and Word Attention-Enhanced Neural Network"

_ijerph, 2020, doi:10.3390/ijerph17051614_

Round 1

Reviewer 1 Report

The paper investigates how deep learning techniques can be used for medical named entity extraction.

The topic is interesting and worth investigating. The paper is very well written and includes a comprehensive literature review. The proposed approach is described with an adequate level of details. The results are significant when compared with several existing approaches.

A discussion should be included highlighting whether a different approach is needed in the context of the Chinese language, compared to English. The authors could also highlight whether the proposed approach can easily be adapted for other languages.

At line 76, the paper use the abbreviation CRF. It is recommended to always explain the abbreviations, when first used, in order to make the paper more accessible to readers that are less familiar with topic.

Author Response

See uploaded file

Reviewer 2 Report

The paper is well written and presents significant applicability.

The authors could consider using a measure regarding the latency of the proposed model and the amount of data used as input.

L. 110. Describe the acronym JNLPBA.

L.115-116. Rewrite this sentence: ” The sentence level features were extracted by CNN, but the sequence features cannot be better obtained.”

L.127-128. There is a lack of information on the text regarding the JNLPBA2004 corpus. Provide more details.

L.190-192. In this part of the text, the authors should give more details regarding the accuracy model.

L. 241-246. The authors should increase the level of detail in this paragraph. It is a crucial part of their work that could be improved.

L. 294-296. Provide more information regarding the reasons to use 19 tags.

L. 396-397. Rewrite this sentence: “We can know that our model out-performed other methods, with precision of 94.16%, recall of 94.72%, F1 of 94.44%.”

L.430-436. Excellent comparison, it also points out when the proposed model has a fast convergence.

Typos:

“dimens ions”

Author Response

See the uploaded file

Reviewer 3 Report

A. Update the following references:

1: 2010, 2:2011, 23: 1997, and 28:2008

B. Explain the acronyms used. e.g. CWE, JNLPBA...

C. Figure 4, need more resolution.

D. Tables format, it is necessary to separate the results. e.g. Table 5

E. It's very important to explain Figure 9. Line black: CWME-BILSTM-CRF

Author Response

See the uploaded file
